# Multiscale Deblurred Feature Extraction Network for Automatic Four-Rod Target Detection in MRTD Measuring Process of Thermal Imagers

**DOI:** 10.3390/s23094542

**Published:** 2023-05-07

**Authors:** Zhenggang Guo, Wei Guan, Haibin Wu

**Affiliations:** School of Mechanical Engineering, Dalian University of Technology of China, Dalian 116024, China; gzg@dlut.edu.cn (Z.G.);

**Keywords:** minimum resolvable temperature difference, neural network, infrared thermal imaging, deep learning

## Abstract

The minimum resolvable temperature difference (MRTD) at which a four-rod target can be resolved is a critical parameter used to assess the comprehensive performance of thermal imaging systems, which is important for technological innovation in military and other fields. Recently, there have been some attempts to use an automatic objective approach based on deep learning to take the place of the classical manual subjective MRTD measurement approach, which is strongly affected by the psychological subjective factors of the experimenter and is limited in accuracy and speed. However, the scale variability of four-rod targets and the low pixels of infrared thermal cameras have turned out to be a challenging problem for automatic MRTD measurement. We propose a multiscale deblurred feature extraction network (MDF-Net), a backbone based on a yolov5 neural network, in an attempt to solve the aforementioned problem. We first present a global attention mechanism (GAM) attention module to represent strong images of the four-rod targets. Next, a Rep VGG module is introduced to decrease the blur. Our experiments show that the proposed method achieves the desired effect and state-of-the-art detection results, which innovatively improve the accuracy of four-rod target detection to 82.3% and thus make it possible for the thermal imagers to see further and to respond faster and more accurately.

## 1. Introduction

Infrared imaging systems are widely used in astronomy, medical diagnosis, industry and the military [1,2]. For example, in military applications, infrared imaging systems are widely used in missile guidance, infrared radar, infrared night vision, infrared reconnaissance, infrared communication, infrared confrontation, regional alert and infrared robots [3,4,5], which are equivalent to the eyes of fighter aircraft, used to collect and sense air combat information and to receive more accurate feedback on the movement status of airborne enemy aircraft, which is a decisive factor in determining the success of military operations. In the medical field, infrared thermal imagers detect infrared light emitted by the body to visualize changes in the body, which are usually applied in early diagnosis, auxiliary diagnosis, clinical curative effect evaluation, traditional Chinese medicine and acupuncture. In recent years, infrared technology has been successfully applied to breast cancer, diabetes, and peripheral vascular disease diagnosis, as well as inspections in the fields of gynecology, renal transplantation and dermatology and related studies [6,7]. Last but not least, infrared thermal image technology has extensive applications in the industrial field, such as in state monitoring and thermal fault diagnosis of the equipment in a power field [8], because it can be tested in the case of no contact and if there is no damage. However, as an important performance index of infrared imaging systems, MRTD is not high enough. And it is well-known that it can only be measured precisely using fabricated measuring equipment. For this reason, it is necessary to build and study a set of more sensitive and precise MRTD testing theories and measuring equipment for infrared imaging systems. Nevertheless, previous studies on MRTD testing methods have focused on manual subjective measuring methods with poor stability and consistency, while less attention has been paid to automatic MRTD measurement using artificial neural networks in deep learning areas. Although some scholars have applied some simple neural network algorithms to the automated measurement of MRTD, there still seem to be some shortcomings in terms of the accuracy and stability of the measurements. For this reason, we will study an algorithm for four-rod target detection based on an artificial neural network to measure MRTD automatically, which is important for technological innovation in many military and civilian engineering fields.

The MRTD equals the minimum temperature difference of the target and background when the four-rod target can be just resolved at a certain spatial frequency of the four-rod target. The four-rod target is a target covered by high-emissivity black coatings with four striped rectangular holes at a 7:1 fixed ratio as shown in Figure 1 below, and the spatial frequency is just the width of a striped rectangular hole. The graph of the MRTD against the spatial frequency is used to characterize the performance of the thermal imaging equipment.

In the past, the classical MRTD measurement method, which is based on manual subjective detection, has been widely accepted but demonstrates several problems, including high labor cost and low efficiency, and the results are too influenced by the psychological factors of the individual to be convincing [9,10].

The development of deep learning makes intelligent MRTD measurement a reality, because the neural network can study to classify and infer like an operator through extensive training. Therefore, the neural network can divide the four-rod target into a distinguishable target, indistinguishable target and threshold target without labor cost, individual difference and working fatigue, especially when assessing quantities of infrared imaging equipment. The studies in [11,12] demonstrated observational inconsistency of a four-rod target between different individuals during classical subjective MRTD measurement. Consequently, it is an urgent challenge to facilitate objective four-rod target detection with higher accuracy, faster detection speed and automation of MRTD measurement.

To address these challenges, we propose a multiscale deblurred feature extraction network (MDF-Net) for four-rod detection in infrared imaging. To resolve the multiscale problem in infrared–MRTD images, a global attention mechanism (GAM) attention module is designed to magnify salient cross-dimension receptive regions and augment data scaling capability and robustness. Secondly, we propose RepVGG, a simple architecture with a stack of 3 × 3 conv and ReLU, which is fast, elegant and simple.

The purpose of our study is to improve the accuracy of neural networks applied to four-rod target classification, thus making the MRTD measurement calibration process more automated and accurate. To a certain extent, more automated MRTD measurement can lead to faster and cheaper production of thermal imaging cameras, and more accurate MRTD measurement can lay the foundation for iterations of thermal imaging cameras with smaller MRTD, which means that the thermal imagers will be able to identify the target at a smaller target background temperature difference, thus improving the performance of the thermal imagers. Our contributions can be summarized as follows:(1)We contribute a dataset of 991 four-rod target photos and make it open-source.(2)We demonstrate the effectiveness of a multiscale deblurred feature extraction network (MDF-Net) in four-rod target detection.(3)We propose the GAM attention module and RepVGG module to lower the influence of the scale variability of the four-rod targets and the low pixel resolution of the photos.

### 1.1. Related Work

#### MRTD Measurement Development

Because of the drawbacks of traditional MRTD measurement methods, other alternative methods have been pursued by some researchers. The studies in [13,14] discussed the possibilities for objective measurements and alternative MRTD measurement methods. Some researchers [15,16,17] used an indirect method to evaluate the MRTD by computing the modulation transfer function and the noise equivalent temperature difference (NETD) of the thermal imager. Furthermore, a mathematical model [18] for numerical estimation of the MRTD has been proposed, while other scientists have attempted to use indirect methods to generate the four-rod targets using computer simulations [19] or a controlled blackbody [20]. The studies in [21,22] proposed a triangle orientation discrimination threshold (TOD) model, minimum temperature difference perceived (MTDP) and thermal range model (TRM3) to replace the MRTD as the performance parameter of infrared imagers. Recently, efforts to implement deep learning into four-rod target detection of MRTD measurement have been pursued. Sun [23] and Ma used a frame grabber to extract the feature vector of a four-rod target based on a Back-Propagation Network for MRTD measurement. Liwei Xu [24] adopted a BP neural network, which optimized both the measurement process and algorithm processing, meeting the MRTD measurement requirements. Rong [25] applied a CNN neural network to extract features for four-rod target image classification problem by optimizing the convolution layer size, changing the activation function and adjusting the network structure.

### 1.2. Multiscale Detection Methods in Deep Learning

Multiscale detection has seen about four iterations since its initial proposal [26]. To begin with, researchers started to solve the problem by building “feature pyramid + sliding windows”, which frequently glide a fixed-size detection window over the image, paying little attention to “different aspect ratios”. From 2004 onwards, many detectors were built based on this paradigm, including the HOG detector [27], DPM [28] and Exemplar SVM [29]. After that, some researchers started to detect using object proposals, which refer to a group of class-agnostic reference boxes that are likely to contain any objects to avoid an exhaustive sliding window search across an image, with some famous detectors such as RCNN [30], SPP [31] Net, Fast RCNN [32] and Faster RCNN [33]. Subsequently, researchers used deep regression to solve multiscale problems, which directly predict the coordinates of a bounding box based on the deep learning features, such as DNN [34] Net and YOLO [35]. Currently, multi-reference detection is the most used method for multi-scale detection, which first define a set of references (anchors, including boxes and points) at every location of an image, and then predict the detection box based on these references. Relevant significant nets include Faster-RCNN [33], SSD [36], FCOS [37], YOLOV4 [38] and YOLO V5.

Inspired by the perfect performance of YOLOV5, we adopted its architecture in this work to demonstrate the effect of the proposed multiscale deblurred feature extraction network (MDF-Net) in four-rod target detection.

### 1.3. Four-Rod Target Detection in Infrared Thermal Imaging Systems

At first, this paper proposed training the BP neural network to read the four-rod target image by extracting the four-rod target image’s feature values as input. The limitation of this method is that obtaining the feature value representing the resolution level of the entire four-rod target image is difficult. Because of its advantages in terms of algorithm and structure, CNN neural networks play an important role in image recognition. Based on a CNN neural network, this paper [25] proposes a new MRTD objective test method. Additionally, Youhui Tao [39] et al. proposed a concealed multiscale feature extraction network for automatic four-rod target detection which can effectively address the challenges in infrared–MRTD images.

## 2. Proposed Method

### 2.1. Infrared Four-Rod Target Dataset

#### 2.1.1. Experimental System Setup

As shown in Figure 2, we set the MRTD measurement system to acquire the four-rod target photos and to lay the foundation for an end-to-end MRTD measurement system online for future application.

The set experiment equipment mainly includes a collimator, optical platform, differential blackbody, four-rod targets with different spatial frequency and computer processing software.

First, to provide a temperature difference signal, we adopted a differential blackbody which can generate 5–50 °C long infrared waves with 59 mm diameter radiation surface as the radiation source. Then, four-rod targets with a series of spatial frequency dependent on the thermal imager to be detected were set on the radiation surface of the blackbody. According to our chosen thermal imager, the width of the bar was set at 1, 0.6, 0.35, 0.3, 0.25, 0.21, 0.18, 0.125 and 0.09 (mm), and the diameter of the target was set at 59 mm, the same as the blackbody. The object temperature going through the rectangle hole of the four-rod target from the blackbody could be changed as we wanted, while the four-rod target itself was maintained at the room temperature of about 20 °C because of the black coating with high emissivity at 0.98 on the surface of the four-rod target. Therefore, the controllable temperature difference between the target and the background can be simulated for the aforementioned reason. However, we still need a collimator to imitate the real work situation of the infrared thermal imaging equipment where the imaging object goal is infinitely far in aerial combat. Hence, parallel light from infinity was simulated by using a reflective off-axis collimator in our study. Through the collimator, the thermal imager on the optical platform can generate images of the four-rod target, which can be sent to the computer processing software for further applications such as data collection, storage, processing and detection.

#### 2.1.2. Dataset Acquisition Process

The data acquisition process can be divided into the following steps:1.Measure the temperature of our experiment chamber as T and maintain it to ensure the accuracy of the test results, put the infrared imager on the optical platform stably, adjust the temperature of the blackbody to T + 10 K and install a four-rod target with fixed spatial frequency on the radiation surface of the blackbody. The four-rod target can then be distinguished clearly on the thermal imager.2.Gradually decrease the temperature of the blackbody by 0.1 K until the temperature difference between the target and the background become negative. Photos of the four-rod target are taken by the thermal imager at each temperature point. Data acquisition ends when the four-rod target can be distinguished clearly in negative temperature difference.3.Invite a group of testers to mark labels for the data; the category labels are divided into “Y”, “N” and “T”, which represent distinguishable targets, totally indistinguishable targets and targets on the margin of distinguishable and indistinguishable, respectively. Then, we finally obtain the GW-MRTD dataset.

### 2.2. Proposed Neural Network

From the last chapter, we can conclude that the character of the dataset has two aspects. First, there are four or five targets with differential spatial frequency which require the measurement network to extract feature maps from different receptive fields. Second, when the net detects the threshold target, the small temperature difference leads to the high similarity of the target and the background, which increases the difficulty of detection.

To solve the aforementioned problems, in this chapter, we introduce the overall network framework containing the multiscale deblurred feature extraction network (MDF-Net) based on YOLOV5. Furthermore, the GAM module and the Rep VGG module are presented subsequently.

#### 2.2.1. Overall Network Framework

As shown in Figure 3 below, the overall network framework concludes the traditional detection head of the YOLOV5 and the backbone MDF-Net proposed, which mainly consists of the focus, bottleneck CSP module, spatial pyramid pooling (SPP), Rep VGG module and GAM attention module.

To begin with, the input images first go through the backbone for feature extraction: the data are divided into four parts in the focus module, where every piece of data is made equivalent to two-times-lower sampling and concatenated together in the channel dimension, and then convolution operations proceed. There are two kinds of convolution modules: the yellow module consists of feature maps convoluted sequentially, batch normalization and Leaky ReLU activation function, where the height and the width of the feature maps are halved to extract features and perform downsampling when conducting a convolution operation because the stride is two; the other orange module cannot change the size of feature maps because the stride is one.

Additionally, the network adopts the Cross Stage Partial Networks to extract abundant information features from the input images. The CSP module can integrate the gradient information into the feature maps to decrease the number of parameters and FLOPS, which not only minimize the size of the model but also ensure the inference speed and accuracy, thus solving the problem of gradient information repeating when optimizing the network in the backbones of many other large convolutional neural network frameworks.

The spatial pyramid pooling module (SPP) is used to perform the 5 × 5, 9 × 9, 13 × 13 pooling operation in the end of the backbone, which will not change the size of the feature maps because of the padding, and the three aforementioned feature maps concatenate with the fourth feature map without pooling and proceed into the next module. The SPP module can expand the receptive field and extract the most significant contextual features without decreasing the inference speed.

Secondly, the data come to the detection head of the YOLOV5: the detection head of the net adopts the path aggregation net module (PANet) to generate the feature pyramid, which can augment the detection performance to differential scaling size targets so as to detect the same target in different size and scales. Based on Mask R-CNN and FPN framework, the feature extractor of PANet adopts a bottom-up path FPN structure to improve the dissemination of the lower features. The feature mappings of the previous stage are processed with 3 × 3 convolutional layers, concatenated and added to the feature maps at the same stage, providing information for the next stage input. At the same time, the adaptive feature pooling recovers the destroyed information path between all the feature layers and aggregates every candidate region on every feature layers.

The Rep VGG and GAM module will be discussed in the next section.

#### 2.2.2. Global Attention Mechanism (GAM) Attention Module

The input of a neural network is generally a multidimensional array (batch size × channels × height × width) converted from color pictures, where the channels represent the channel dimension and the width and height represent the spatial dimensions. During training, the convolution operation can mix channel and spatial information to extract feature information. Just like human eyes can focus, neural networks can also selectively focus on certain information in an image, which is exactly what the attention mechanism does during training. Although the convolutional block attention module (CBAM) has been proposed to solve this problem, which places the channel and spatial attention operation sequentially, it ignores the cross interaction between the channel and the space, thus losing the cross-dimensional information.

However, the global attention mechanism (GAM) module does not address the problem which the CBAM encounters by reducing the information, decreasing and amplifying the global dimension interaction features; the global attention mechanism (GAM) moduleis called “global” not only because the attention operations are applied on three dimensions, including the channel, spatial height and spatial width simultaneously rather than sequentially each time, but also it takes the interactions between dimensions into consideration. As shown in Figure 4, the GAM redesigns the submodule derived from the sequential channel–spatial attention mechanism in CBAM.

The process can be expressed in the following Formulas (1) and (2). With the given input feature mapping $F_{1}\in R^{C*H*W}$, the intermediate state $F_{2}$ and the output $F_{3}$ can be defined as:(1)$$F_{2}=M_{C}\left(F_{1}\right)⊗F_{1}$$
(2)$$F_{3}=M_{C}\left(F_{2}\right)⊗F_{2}$$

The added submodules mainly comprise two parts. One is the channel attention, as shown in Figure 5, which firstly adopts three-dimensional permutation to retain the three-dimension feature and then magnifies the relativity of the cross-dimension channel and space through two layers of Multi-Layer Perceptron.

The other part is spatial attention, as shown in Figure 6, which first adopts two convolutional layers to integrate the spatial information for the centralization of the spatial information. Meanwhile, the maximum pooling is removed to obtain further retainment of the feature mapping due to the fact that the maximum will actually decrease the information. Additionally, the data are randomly divided into several groups sent to random convolution channels, which can effectively prevent the augmentation of the parameters.

#### 2.2.3. Rep VGG Module

Multi-branch convolutional network architectures have been used since the proposition of the Resnet and inception, while the classical easy and unbranched “VGG” style modules are rarely used. The REP VGG, however, revives the “VGG”-style modules through structural re-parameterization, which concludes training a multi-branch model, equivalently converting the multi-branch model into a single-socket model and deploying the single-socket model.

As shown in Figure 7, the RepVGG first accumulates more than 20 layers and then divides them into 5 stages, where the first layer of every stage concludes downsampling with 2 strides and ReLU as the activation function. An intact RepVGG block concludes the 1 × 1 convolution branch and the identical mapping branch added for every 3 × 3 convolution layer when training. As a result, we make equivalent conversions of the model and obtain the deployment model.

Here, we can see the essence of “structural re-parameterization”: the training structure corresponds to a set of parameters, while the structure we want in inference corresponds to another set of parameters. Then, as long as we can equivalently convert the former’s parameters to the latter’s parameters, we can equivalently convert the former’s structure to the latter’s structure.

## 3. Experiments and Discussion

In this section, we verify the effectiveness of the GAM attention module and the Rep VGG module through our ablation experiment and compare the performance of our proposed MDF-Net with other algorithms.

(1)Dataset: A dataset of 991 images with 1280 × 1280 pixels was used to train our detection model. The whole GW-MRTD dataset (991 images) can be divided into a training set (921 images), a valid set (50 images) and a test set (20 images). To begin with, we applied pre-processing to the original data, including auto-orient and stretching to a size of 640 × 640. To improve the generalization of the net and mitigate the overfitting, some data augmentations were adopted, such as cropping from 0% minimum zoom to 10% maximum zoom, noise up to 5% of pixels and mosaic. Next, we generated coco-formatted datasets for training using annotation tools. Finally, the backbone of the yolov5 framework was pretrained on the coco128 dataset and trained on our GW-MRTD dataset for 100 epochs.(2)Experimental configuration: We adopted Pytorch as the framework. An Intel Core i7-8700 CPU and an NVIDIA RTX3060 with 128 GB of memory made up the computer’s processor setup. Ubuntu 18.04 was used as the default operating system.(3)Detection criteria: Precision (P), recall (R) and achieved mean average precision (mAP) are three commonly used indices that are used to objectively evaluate the effectiveness of defect detection methods. P is the proportion of “true” samples among all “true” samples as determined by the system. R is the proportion of “true” samples among all true samples. mAP is the average of AP for each category, and AP reflects the detection accuracy of a particular category. We further fine-tuned the hyperparameters and finally set the batch size at 16 and the learning rate at 0.01. After the process, MAP_0.5 (mean average precision when IOU set at 0.5) was employed as the evaluation metric, which equals the average of all categories of the area under the precision–recall curve of a certain category.

### 3.1. Ablation Experiment

To assess the performance of the proposed modules separately, we designed a comparative experiment on the framework based on the control variable method. The performance on images is illustrated in Figure 8 below.

As shown in Figure 8, the input images predicted the four-rod target separately through the GAM module only, RepVGG module only and both the GAM module and RepVGG module. The GAM module is designed to capture robust representation of multiscale targets in different scenarios. Table 1 shows that the mAP metric improved significantly with the insertion of the GAM module in the framework, increasing by 2.7%. As demonstrated in Figure 8b, the feature maps generated from the GAM module exhibited sensitivity to the multiscale target region while retaining detailed information of the overall scene, but the results show some false detection. The RepVGG module distinguishes the four-rod targets from the background by encoding local and global salient context. It can be observed from Table 1 that the RepVGG module improved the mAP metric by 3.4% over the baseline. As illustrated in Figure 8c, the feature maps processed by the RepVGG module maintained high consistency throughout the whole scenario, showing significant differences only in the target regions, but the results still show some false detection.

To explore the full potential of the modules, we also implemented different connection approaches between the modules to evaluate the impact. As shown in Table 1 and Figure 8d, both connection methods obtained good performance; specifically, both the connection approaches outperformed the single-module approaches in all categories of AP metrics and were 8.6% higher in the mAP metric.

To further quantify the model effect, we used MAP_0.5 as an evaluation index. The results are shown in Table 1 below.

As shown in Table 1, the MAP_0.5 of the GAM module increased by 2.7%, while the MAP_0.5 of the RepVGG module increased by 3.4%. Further, the MAP_0.5 of both the GAM module and the RepVGG module significantly increased by 8.6%, which demonstrated the effect of our proposed MDF-Net in the four-rod target detection experiment.

### 3.2. Comparison with Other Nets

To verify the performance of our proposed MDF-Net, we compared our method with other state-of-the-art (SOTA) methods, including RCNN, YOLOV3 and YOLOV4, on the GW-MRTD dataset. We used the same training and testing procedure as our proposed method for other SOTA detectors to present a fair performance evaluation. All of the other SOTA detectors were finetuned on GW-MRTD using the same official pretrained model, with only the hyperparameters changed.

As is illustrated in Table 2, our method outperformed other detection nets not only in the MAP but also in the AP of every category (Y represents distinguishable, N represents indistinguishable and T represents threshold).

It can be seen that the recognition performance at 72.4% was good even for the simplest RCNN neural network, which indicates that the neural network can perform the four-rod target detection task well. Meanwhile, the data show that the YOLO series neural networks matched this detection task and context very well, increasing by about 5% compared with RCNN. Finally, we can find that our proposed yolov5-based MDF-Net significantly increased by 1.2% over the original yolov5 neural network.

On the IR-MRTD test set, Figure 9 shows some prediction results of our proposed Net and RCNN Net as a comparation. It can be clearly seen that all the targets of various categories can be precisely identified and localized. Our approach enables accurate detection and classification of even very small targets and performed better than the RCNN Net on the left, which is essential for infrared imaging equipment performance assessment.

The limitations of the proposed approach mainly focus on two aspects. One is that the volume of the dataset was not large enough, with only 991 photos of the four-rod target, which is because some of the experimental instruments were borrowed from the Shenyang Automation Institute of the Chinese Academy of Sciences, and more time could not be scheduled as the Institute also needed to use the instruments. The other limitations were that the performance of our GPU device was not good enough and the training process was time-consuming; therefore, we could refine the hyperparameters and optimize the algorithms in the future.

## 4. Conclusions

In this study, we proposed a multiscale deblurred feature extraction network (MDF-Net), a backbone based on a yolov5 neural network, in an attempt to improve the efficiency and accuracy of objective MRTD detection in thermal imaging systems. Our experiments show that the proposed method achieved the desired effect and excellent detection results. The main conclusions are as follows:1.The GAM module and the RepVGG module both play an important role in the detection of the four-rod target, increasing by a mAP at 2.7% and 3.4% separately.2.The proposed MDF-Net based on YoloV5 achieved a mAP of 82.3% on the test photos of the four-rod target, indicating that the model proposed can identify and classify the four-rod target into “Y”,” N” and “T”. The research results suggest that training an artificial neural network to measure the MRTD automatically is promising.

The great detection performance of the four-rod target provides a more sensitive and precise MRTD testing theory and lays a foundation for infrared imaging systems, which can be applied in the parameter (MRTD) calibration of thermal imagers in production as well as save labor costs during the production of thermal imaging cameras. Once the manufactured thermal imagers are put into use, due to the increased accuracy of the automated MRTD measurement, the MRTD of the thermal imagers will be smaller, which means that the thermal imagers will be able to identify the target at a smaller target background temperature difference, thus improving the performance of the thermal imagers.

## Figures and Tables

**Figure 1 sensors-23-04542-f001:**
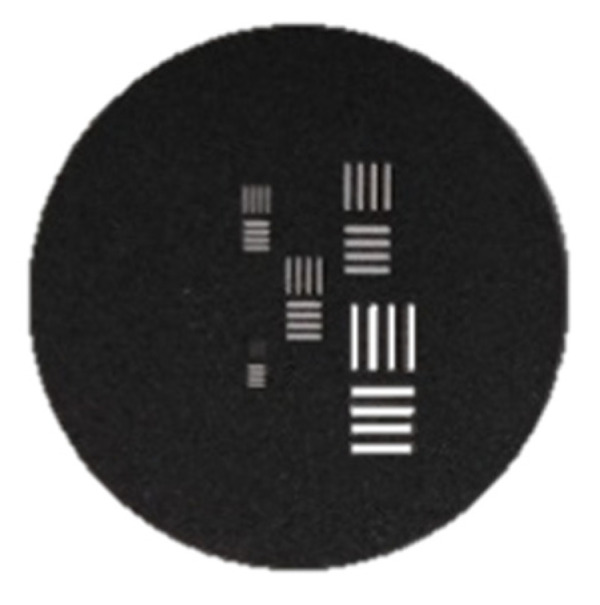
Four-rod target.

**Figure 2 sensors-23-04542-f002:**
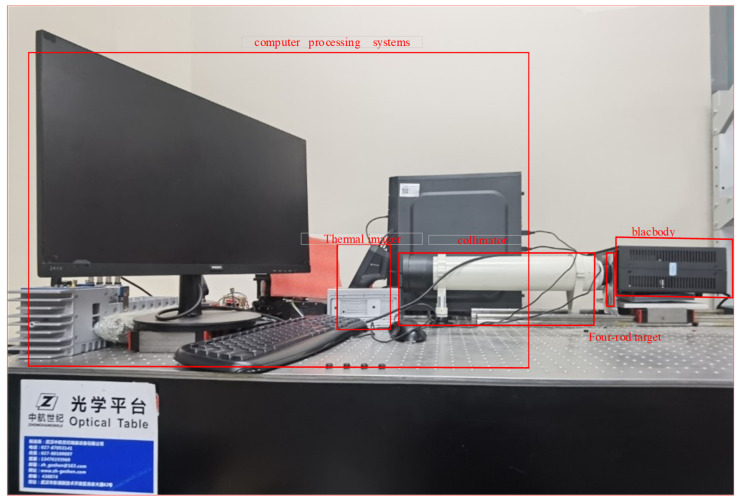
MRTD measurement system.

**Figure 3 sensors-23-04542-f003:**
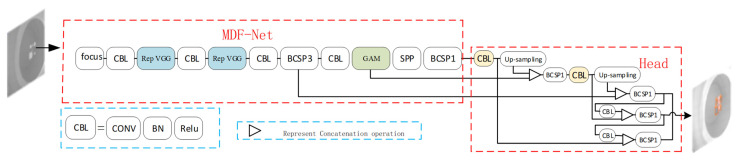
Overall network framework.

**Figure 4 sensors-23-04542-f004:**
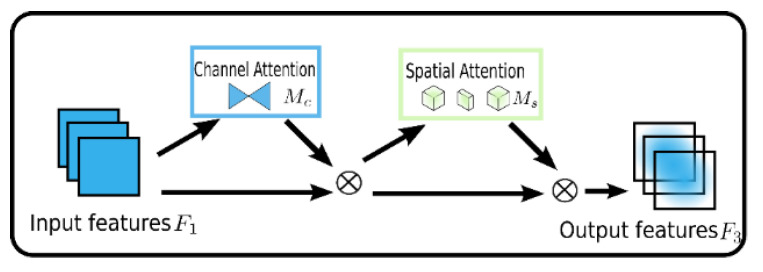
Global attention mechanism (GAM) attention module.

**Figure 5 sensors-23-04542-f005:**
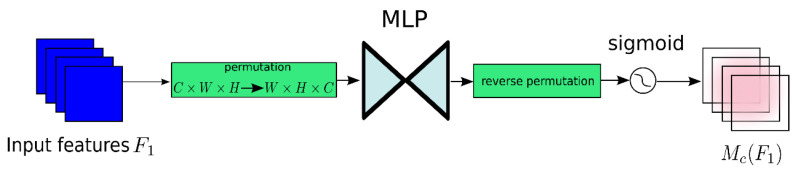
Channel attention submodule.

**Figure 6 sensors-23-04542-f006:**
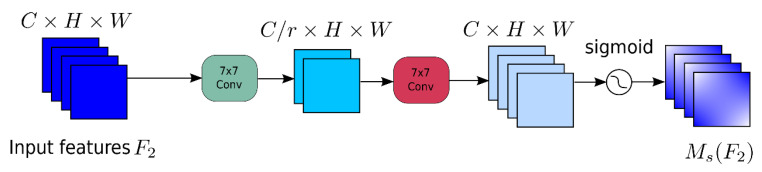
Spatial attention submodule.

**Figure 7 sensors-23-04542-f007:**
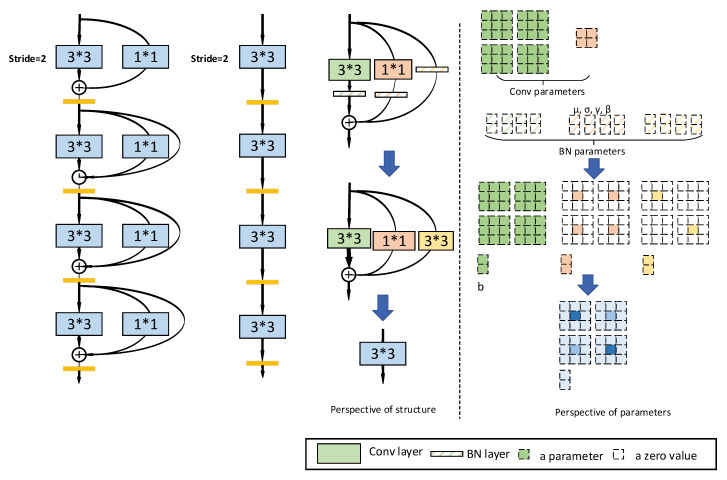
Rep VGG module.

**Figure 8 sensors-23-04542-f008:**
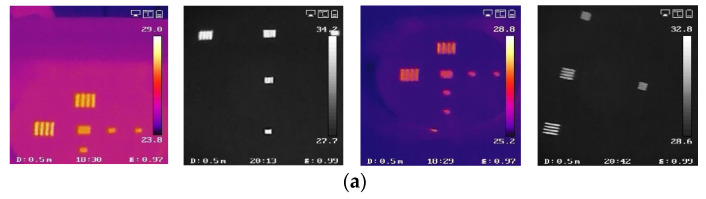
Separate effectiveness of different situations: (**a**) the input images; (**b**) the output images of the GAM module only; (**c**) the output images of the RepVGG module only; (**d**) the output of both the GAM module and RepVGG module.

**Figure 9 sensors-23-04542-f009:**
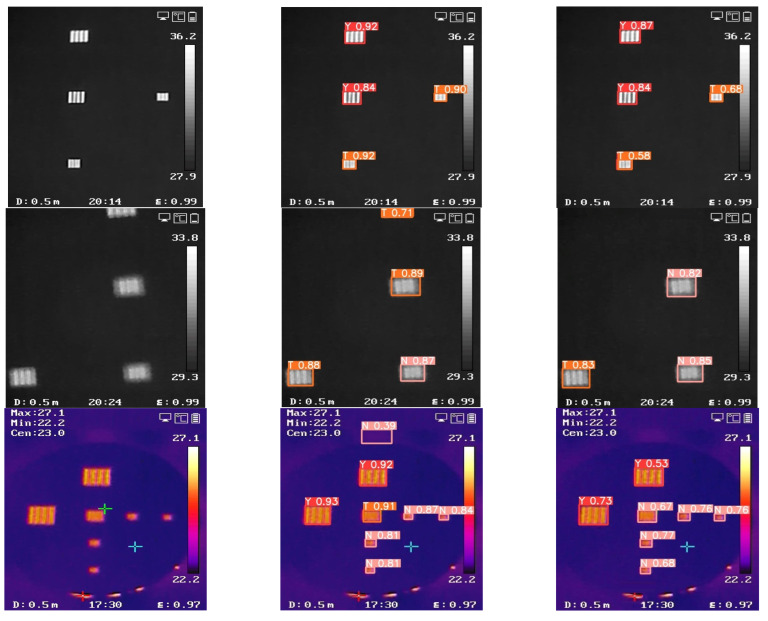
Visualization of detect results of different nets: (**a**) original images; (**b**) RCNN; (**c**) our method.

**Table 1 sensors-23-04542-t001:** MAP results of the ablation experiment.

GAM	RepVGG	MAP_0.5 (%)
0	0	73.7
0	1	78.9
1	0	79.6
1	1	82.3

**Table 2 sensors-23-04542-t002:** MAP results of the comparative nets.

Nets	MAP_0.5 (%)	AP _Y_ (%)	AP _N_ (%)	AP _T_ (%)
RCNN	72.4	89.7	79.4	70.3
YOLOV3	77.5	92.1	80.1	69.8
YOLOV4	79.6	94.7	81.9	72.1
YOLOV5	81.1	95.2	80.9	72.4
Our method	82.3	97.6	83.5	73.8

## Data Availability

We published our dataset at https://github.com/Guanwei6/four-rod-target-dataset/tree/main/4-bar-%20classification.v4i.yolov5pytorch, accessed on 1 May 2023.

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
