# Peer review of "Multiscale Deblurred Feature Extraction Network for Automatic Four-Rod Target Detection in MRTD Measuring Process of Thermal Imagers"

_sensors, 2023, doi:10.3390/s23094542_

Round 1

Reviewer 1 Report

This report (ID: sensors-2370084) entitled “Multiscale deblurred feature extraction network for automatic four-shot target detection” by research group Dr. Wei Guan. This article is interesting and can be important contribution to the scientific literature. However, the authors have to address formal flaws and strengthen the presentation and clarity. Reviewer suggest minor revision is needed before publication in journal. Some specific comments on the manuscript:

1. Originality

State the novelty of the article in the abstract, introduction sections.

2. Abstract

Line 14: authors state for technological innovation in the military field and other fields, state the other fields and explore applications. Validate other field applications by experimental and literature evidence.

Define and provide the full form of Rep VGG module.

3. Methodology:

*3. Proposed Method: 3.2. global: why it is called global.

4. Experiments: validate its application in other fields.

4. Results and discussion:

*Scientific clarity, mechanisms and illustration of results can be improved.  

*Specify the limitations of methods and materials at the end of the section.

5. Conclusion

*Focus on the scope (such as future implications, sustainability, cost-effectiveness, and improvement in performance)

Need to improve the text 

Reviewer 2 Report

1. The correlation between the article title and content is inadequate. The title proposes "Multi-Scale Deblurring Feature Extraction Network", while the content is about the measurement method of the key parameter MRTD. It is recommended to modify the title.

2. The logic in the abstract is unclear and does not reflect the superiority of the proposed method. Moreover, the statement "achieves state-of-art detection results compared with other existing methods" is too absolute. It is suggested to rewrite the abstract.

3.   The introduction does not include relevant research or literature to prove the feasibility of deep learning in measuring MRTD. It is recommended to include related literature as support in the introduction.

4. Deep learning methods require a large amount of image data to train the model, but the dataset used in this study only contains 991 target photos. Will this data quantity affect the model training?

5. The research method is based on improvement of YoloV5, and the accuracy comparison with the original YoloV5 model can be added in the model comparison part.

6.  The discussion in the conclusion section of the research method is too simple and lacks a clear analysis. Moreover, the content is no different from that in the abstract. It is suggested to rewrite this part.

The English is generally OK, with a few grammatical errors

Reviewer 3 Report

The manuscript is devoted to the development of an automated method of target search in thermal imaging sights.

The authors have done a lot of work, but the Manuscript itself needs serious structural revision. 

1. The annotation needs to be improved, since most of it is devoted to the relevance and formulation of the problem, and not to the results.

2. The manuscript must be structured according to the traditional IMRAD format, otherwise it is difficult for readers to perceive.

3. The design of links and References does not meet the MDPI requirements.

4. What is the specific purpose of the study? It should be formulated at the end of the Introduction.
